# Understanding the Clinical Impact of MUC5AC Expression on Pancreatic Ductal Adenocarcinoma

**DOI:** 10.3390/cancers13123059

**Published:** 2021-06-19

**Authors:** Ashish Manne, Ashwini Esnakula, Laith Abushahin, Allan Tsung

**Affiliations:** 1Department of Internal Medicine, Division of Medical Oncology at the Arthur G. James Cancer Hospital and Richard J. Solove Research Institute, The Ohio State University Comprehensive Cancer Center, Columbus, OH 43210, USA; Laith.Abushahin@osumc.edu; 2Department of Pathology, The Ohio State University Wexner Medical Center, Columbus, OH 43210, USA; Ashwini.Esnakula@osumc.edu; 3Department of Surgery, The Ohio State University Wexner Medical Center and James Cancer Hospital and Solove Research Institute, Columbus, OH 43210, USA; Allan.Tsung@osumc.edu

**Keywords:** pancreatic ductal adenocarcinoma, biomarkers, predictors, MUC5AC, NPC1-C, PAM4, CLH2, mucin

## Abstract

**Simple Summary:**

Management of pancreatic cancer is challenging as there are limited treatment options, and most cases are diagnosed at advanced stages. In addition, there are no dependable tests available to predict bad outcomes or treatment responses in current clinical practice. Here, we shed light on the available evidence on mucin, MUC5AC in predicting the outcome of pancreatic cancers. We also discuss variants of MUC5AC believed to have a role in the malignant transformation of pancreatic tissues.

**Abstract:**

Mucin-5AC (MUC5AC) is a heavily glycosylated gel-forming secreted mucin with a reliable prognostic value when detected in multiple malignancies. It is highly prevalent (70%) in PDA and is nonexistent in normal pancreatic tissues. Retrospective studies on PDA tumor tissue (detected by immunohistochemistry or IHC)) have investigated the prognostic value of MUC5AC expression but were equivocal. Some studies associated it with poor outcomes (survival or pathological features such as lymph node disease, vascular/neural invasion in resected tumors), while others have concluded that it is a good prognostic marker. The examination of expression level threshold (5%, 10%, or 25%) and the detected region (apical vs. cytoplasmic) were variable among the studies. The maturation stage and glycoform of MUC5AC detected also differed with the Monoclonal antibody (Mab) employed for IHC. CLH2 detects less mature/less glycosylated versions while 45M1 or 21-1 detect mature/more glycosylated forms. Interestingly, aberrantly glycosylated variants of MUC5AC were detected using lectin assays (Wheat Germ Agglutinin-MUC5AC), and Mabs such as NPC-1C and PAM4 have are more specific to malignant pancreatic tissues. NPC-1C and PAM4 antibody reactive epitopes on MUC5AC are immunogenic and could represent specific changes on the native MUC5AC glycoprotein linked to carcinogenesis. It was never studied to predict treatment response.

## 1. Introduction

Cancer of the exocrine pancreas is a highly lethal malignancy. It is the fourth leading cause of cancer-related death in the United States and second only to colorectal cancer as a cause of digestive cancer-related death [1]. According to recent SEER data, the five-year survival rate is only 10% for all stages and a disappointing 40% for even early stage tumors [2]. The lack of prognostic biomarkers with reliable predictive value has handicapped the effective management of pancreatic ductal adenocarcinoma (PDA). 

Carbohydrate antigen 19-9 (CA 19-9) is a blood-based molecular biomarker frequently used to monitor the treatment response in PDA. However, it has some concerning limitations. CA 19-9 is falsely elevated in benign diseases such as acute cholangitis, cirrhosis, and cholestatic diseases [3]. Alternatively, it is not elevated in patients (approximately 5–10%) without a Lewis antigen [4,5]. Histologic characteristics from resected PDA also provide prognostic value such as size of the primary lesion, histologic grade, neurovascular invasion, the presence of tumor necrosis, depth of the portal vein wall invasion, and margins of resection [6,7,8,9]. Unfortunately, there are no routinely used tissue-based predictive and prognostic biomarkers. Currently in the clinical setting, periodic imaging and serum CA 19-9 are used to monitor response to treatments [10]. Thus, there has been considerable enthusiasm to identify ideal tissue and blood biomarkers in PDA in the last two decades, including mucins, osteopontin, cell-free DNA, stromal markers (hyaluronidase), and epigenetic markers [11]. In this review, we will focus on mucin expression in tissues of PDA and the studies that looked at the impact on the outcome and the studies that described variants of MUC5AC.

## 2. Mucins in PDA

Mucins (MUC proteins) are high-molecular glycoproteins with N-acetyl galactosamine (oligosaccharides) sidechains linked by O-glycosidic linkage to specific amino acid residues that occur in repetitive short stretches (tandem repeats) in the backbone [12]. They form a layer on normal respiratory and digestive tracts and guard the underlying epithelial tissues against inflammation, infection, acid (in GI tract), and other physiological insults [13].

Mucins are vital constituents of pancreatic tissues, and their expression patterns (Table 1) can indicate the underlying aberrations in pancreatic tissue. For example, normal pancreatic tissues do not express MUC4 or MUC5AC. However, MUC1, MUC3, and MUC6 are expressed in all kinds of pancreatic tissues from normal to cancerous. In addition, MUC2 expression is weak in normal pancreatic tissues and is not associated with pre-cancerous lesions such as pancreatic intraepithelial neoplasia (PanIN), while MUC4 is not detected in benign pancreatic diseases and its expression is very low in pre-malignant and malignant pancreatic tissues [14,15,16,17]. In normal pancreatic tissues, transmembrane mucins such as MUC1 regulate the proliferation and differentiation of epithelial cells and suppress malignant transformation [18]. Studies have linked aberrant expression of mucins with uncontrolled cell proliferation, distant metastasis, macrovascular invasion, and chemoresistance [15,19,20,21,22].

Among all the investigated mucins in pancreatic tissues, MUC5AC stands unique, as its mere detection in pancreatic tissues is abnormal and is expressed in high levels in cancerous and pre-cancerous lesions. Correlative studies in the literature show its association with poor prognosis in certain cancers (uterine, ampullary cancer, and cholangiocarcinoma) and undesirable mutations such as KRAS in lung cancer [23,24,25,26]. The impact of MUC5AC expression in PDA on patient outcomes, especially survival and treatment response, is currently unknown. This review aims to understand the current evidence on the clinical significance of MUC5AC expression in PDA-tumor tissues and evaluate its use as a potential molecular biomarker (specifically as predictive, prognostic, monitoring, and response).

## 3. PDA and MUC5AC Expression

MUC5AC is gel-forming mucin expressed in the normal tissues (native or normal MUC5AC) from the stomach (gastric pits), gallbladder, conjunctiva, middle ear, nasopharynx, and bronchial tract, and protects their epithelial surfaces from various insults [27,28,29,30]. De novo expression of native MUC5AC in pancreatic tissues occurs in abnormal tissues ranging from benign diseases (pancreatitis) to precursor lesions (PanIN) to mucinous neoplasms and invasive adenocarcinomas; it is well-established in the literature and is attributed to considerable alterations at the gene expression level [15,31,32,33,34]. Its expression along with KRAS is common in intraductal papillary mucinous neoplasms (IPMN), but there is no clear correlation with its malignant transformation [35]. MUC5AC expression (IHC) is observed in all kinds of IPMN (intestinal and gastric) and helps to distinguish it from serous cystadenoma [36,37]. When used along with CA 19-9, serum MUC5AC is useful in that it helps differentiate PDA from other benign conditions like chronic pancreatitis [38,39]. When MUC5AC is added to CA 19-9, specificity improves from 43% to 83% and sensitivity from 79% to 83% in differentiating cancers from chronic pancreatitis/normal. It can be detected (mRNA expression) in the pancreatic tumor aspiration fluid (along with MUC1) of high-grade PanIN and PDA [40]. In IPMN management, MUC5AC detection in the extracellular vesicles can be a reliable biomarker to predict invasiveness and determine the need for surgery, as demonstrated by Yang et al. [41]. Among the IPMN patients, MUC5AC in the extracellular vesicles help in identifying invasive IPMN from low-grade IMPN with a sensitivity of 82%, specificity of 100%, and a diagnostic accuracy of 96%.

The MUC5AC gene is located within a single 400 kb genomic DNA segment on chromosome 11 in the subtelomeric locus 11p15.5 [42]. On this locus, MUC5AC is clustered with four other mucin genes in a sequence MUC6-MUC2-MUC5AC-MUC5B from telomere to the centromere [43]. MUC5AC plays a role by influencing various critical pathways implicated in the malignant transformation and tumor progression in PDA, and the effects are pathway-specific. It accelerates distant metastasis by (a) enhancing the expression of integrins (α3 and β3), matrix metalloproteinase (MMP) -3, vascular endothelial growth factor (VEGF) -A; (b) activating extracellular-signal-regulated kinase (ERK) and VEGF receptor 1 (VEGFR1)pathways; (c) enabling disruption of E-cadherin/β-catenin axis by Kru¨ppel-like zinc-finger protein GLI1 [44,45]. It protects the neoplastic tissue from host immune responses by inhibiting TNF-related apoptosis-inducing ligand (TRAIL) induced apoptosis and decreasing the production of chemokines (CXCL8) [46,47]. It provides resistance from chemotherapeutic agents such αas gemcitabine by promoting E-cadherin/β-catenin axis activation [48]. We summarized the major pathophysiological mechanisms influenced by MUC5AC and the resultant effects in a pancreatic tissue that promote tumor progression and distant metastasis in Figure 1.

Factors instrumental in regulating MUC5AC expression in pancreatic tissues and promoting hyperplasia or dysplasia are not clear. Forskolin is a root extract from Coleus forskohlii that promotes Cyclic Adenosine Monophosphate, or Cyclic AMP (cAMP) by activating the adenyl cyclase enzyme [49]. Vasoactive intestinal peptide (VIP) -1 receptors are consistently found on pancreatic tumor cells [50]. In vitro studies showed that Forskolin and VIP stimulate the expression and release of MUC5AC antigen in pancreatic cells [51]. Transcription factors such as GL-1 (as explained above), Specificity protein factor -1 (SP-1), and Activator protein factor -1 (AP-1) promote the transcription of MUC5AC genes [45,52]. Mutated GNAS (a gene encoding G protein stimulating α subunit) in IPMN upregulates expression of MUC5AC via mitogen-activated protein kinase (MAPK) and phosphatidylinositol 3-kinase (PI3K) pathways [53]. The factors and mechanisms regulating MUC5AC expression in the lung or nasal epithelial were not included in this paper. There is preclinical evidence that MUC5AC has a role in increasing the cell viability, invasiveness, angiogenesis and distant metastasis. However, more studies are needed to prove its clinical utility [54].

## 4. The Impact of MUC5AC Expression on Pancreatic Cancer Outcomes in Clinical Studies

The role of MUC5AC as a useful biomarker with predictive or prognostic value is not well understood. A comprehensive summary of studies that involved immunohistochemistry (IHC) expression of MUC5AC in pancreatic cancer is illustrated in Table 2 [55,56,57,58,59].

Out of 5 studies listed in Table 2, IHC was performed on tissue microarrays, fine needle aspiration (FNA) samples, and resected tissues. In resected specimens, when the threshold is high (25%), the outcomes are poor, consistent with the data on ampullary cancers [23]. In studies where the study population included resected tumors, locally advanced and metastatic patients, the results were inconclusive [55]. MUC5AC function varies based on histology and differentiation of the tumors. In the study done by Higashi et al., MUC5AC had prognostic value only in advanced-stage cancers (Stage III/IV) but was associated with better outcomes. It should be pointed out that the threshold used in this study was just 10%.

Results of our prior tissue-based IHC study were also inconclusive [60]. We attempted to study the expression pattern of native MUC5AC in PDA from 45 patients (25 patients with resectable tumor and 20 patients with metastatic tumor) using CHL2 Mab. Cytoplasmic and apical MUC5AC expression was identified in 82% and 80% of PDA, respectively. In our cohort, apical MUC5AC expression was different between metastatic and non-metastatic patients and showed prognostic significance in non-metastatic patients (better survival in MUC5AC-positive PDA). On the other hand, cytoplasmic MUC5AC expression had no prognostic value. In patients with resectable PDA, MUC5AC expression (apical) detection correlated with better overall survival.

While there appears to be no conclusive evidence that tissue-based MUC5AC IHC detection is reliable, Yonezawa et al. studied MUC5AC gene expression (mRNA using in situ hybridization) in intrapapillary mucinous neoplasms (IPMN) and PDAs [61]. Among PDAs, only 13% expressed MUC5AC and all were in advanced stage tumors (III or IV). There was a correlation between MUC5AC expression and histological subtype (along with MUC2). Papillary structures in cancer tissues of main ducts, interlobular or intralobular ducts and mucinous neoplasms were MUC5AC positive, while tubular and/or poorly differentiated structures were MUC5AC negative. However, MUC5AC mRNA expression in PDA did not make a difference in survival.

While investigating the reasons for discordant results on MUC5AC expression by IHC on PDA prognosis, there has been a focus on the Mabs used. All the studies listed in Table 2 utilized CHL2 Mab that recognizes the sequence TTSTTSAP within the tandem repeats (backbone) of MUC5AC glycoprotein [62,63]. This antibody also detects native MUC5AC constitutively expressed in normal tissues of the lungs and gastrointestinal tract. Other commonly used Mabs, 45-M1 and 2-11M1, also detect native MUC5AC, but the 45M1 Mab recognizes cysteine-rich subdomains of class-2 (Cys2 and Cys4) in the N-terminal domains, and 21M1 Mab reacts with the C-terminal region of MUC5AC [34,64]. These Mabs engage with different epitopes and help identify different variants of native MUC5AC [51,65,66,67]. While CLH2 Mab reacts to immature and less glycosylated mucins primarily concentrated in the perinuclear region, the 45M1 and 21M1 Mabs recognize mature and fully glycosylated forms of MUC5AC concentrated in the cytoplasm or extracellularly (secreted). The prognostic values of either 45M1 or 21M1 have not been studied in PDA or compared with CLH2 Mab.

## 5. Glycosylation of MUC5AC and Pancreatic Cancer

As studies continue to identify changes in MUC5AC that may be associated with malignant transformation of pancreatic tissues, there is now available evidence on variants of MUC5AC secondary to glycosylation. Glycosylation refers to the addition of a carbohydrate-based molecule (glycan) to a protein [68]. Glycosylation of mucins and their role in pancreatic cancer is well-documented in the available literature [34,69,70]. Depending on the site and the number of glycans involved, multiple isoforms (glycoforms or glycan variants) can be produced from a single mucin. Glycan variants of MUC5AC such as Sialyl-Lewisa (SLea), Sialyl-Lewisx (SLex), STn, and Thomsen-Friedenreich (T) (STn/T/SLea/SLex-MUC5AC) have been associated with mucinous adenocarcinomas from the stomach, ampulla of Vater, colon, lung, breast & ovary [71].

In pancreatic cell lines, pro-inflammatory stimuli such as oxidative stress and treatment with the cytokines interferon (IFN) gamma, interleukin IL-1alpha, and TNF-α can induce changes in glycosylation of MUC5AC (along with MUC1 and MUC16) that promote carcinogenesis [69]. Pan et al. reported increased aberrant N-glycosylation levels (using quantitative glycoproteomics) in pancreatic cancers compared to normal tissues or chronic pancreatitis [70]. Native MUC5AC seems to be an early ‘event’ in pre-cancerous pancreatic tissues (such as PanIN), whereas glycosylation with Sialyl-Tn (STn) glycan (non-specific to MUC5AC) is essential to initiate the malignant transformation [34]. Similarly, Yue et al. reported commonly increased glycan alterations in the sera of pancreatic cancer patients such as Thomsen-Friedenreich antigen, fucose, and Lewis antigens [72].

## 6. Wheat Germ Agglutinin-MUC5AC Variant

Haab et al. explored detecting cancer-specific glycan variants of mucins like MUC1, MUC5AC, and MUC16 using the antibody-lectin sandwich microarray method in pancreatic cystic fluid samples [73]. They hoped to identify the biomarker that would differentiate mucin-producing cystic tumors from benign cystic lesions. The MUC5AC variant detected using lectin, Wheat Germ Agglutinin (WGA) (WGA-MUC5AC), was found to be useful with a sensitivity of 76% and specificity of 80%. In this study, other glycan variants of MUC5AC and the variants of other mucins could not distinguish a malignant pancreatic lesion from a benign one. Correlative studies are needed to determine the impact of WGA-MUC5AC on PDA outcomes.

## 7. Niemann Pick C1 (NPC1) Antibody Reactive MUC5AC Variant

In 1970, Hollinshead et al. examined colon/rectal cancer patients exposed to soluble fractions of their autologous tumor cell and normal cell membranes (by dermal injection) [74]. Delayed hypersensitivity reaction was observed in 90% (17/19) of subjects with tumor cell membrane extracts. Normal tumor cell extracts did not elicit a reaction in any of them (0/19). The test was positive (>5 mm within 48 h) in 90% (17/19) of the subjects injected with tumor membranes and negative in all the subjects for normal cell membrane extracts (100%). The antigens responsible for the reaction were assumed to be present only in tumor cell membranes but not in normal cells and were distinct to carcinoembryonic antigen (CEA). This study paved the way for a phase I vaccine trial in 1984, where 22 colon cancer patients received a vaccine with tumor-associated antigen (TAA), prepared from human colon carcinoma cells (surgical specimens) from 70 different donors [75]. The monoclonal antibodies elicited with TAA, also known as NPC1, were used to test vaccines in other cancers like melanoma, but the results were disappointing [76]. Importantly, the target for the NPC-1 (TAA) was later identified as an aberrantly glycosylated MUC5AC variant (cancer-specific MUC5AC) that is different from native MUC5AC [77,78].

Two main studies have used recombinant chimeric NPC1 antibodies prepared in Chinese hamster ovary cells (NPC-1C) to identify cancer-specific MUC5AC. In the first study by Luka et al., the exclusivity of cancer-specific MUC5AC to colon and pancreatic cancers was confirmed in tumor cell lines, tumor, normal tissues, and in the serum of cancer patients. In the first step, supernatant fluid from various cancer cell lines (pancreas, colon, lung, squamous cell carcinoma) was admixed with NPC-1C Mab in a competitive ELISA assay (using MUC5AC antigen-coated microtiter plates). The authors hypothesized that cancer-specific MUC5AC from the tumor cell lines in the supernatant fluid competes with NPC-1C Mab. However, such inhibition was observed only with pancreatic and colon tumor cell lines and was not seen with lung cancer cell lines (which express native MUC5AC) or squamous cell lines. Next, they used NPC1C Mab to stain the tumor tissues from the colon, pancreas, lung, and normal tissues from the colon and pancreas. As expected, tumor tissues from the colon and pancreas expressed tumor-specific MUC5AC while the normal colon/pancreas and lung cancer tissues did not. Interestingly, lung cancer tissues stained with 45M1 Mab reacts to native MUC5AC but not to NPC-1C Mab. Authors report 79% (30/38) of tumor tissues stained positive. Serum cancer-specific MUC5AC (NPC-1C antigen) detection by sandwich ELISA colon and pancreatic cancer patients also has reasonable sensitivity and specificity in distinguishing cancer from controls [77]. Approximately 80% of cancer patients had higher levels of NPC-1C-reactive MUC5AC. However, serial antigen testing did not give valuable information on monitoring the response after initiating treatment. Interestingly, the serum NPC1C antigen’s average was not different between colon and pancreatic cancer patients.

In the second study by Patel et al., NPC-1C Mab’s binding capacity to tumor cell lines (pancreas and colon) was investigated by flow cytometry. They also investigated the immunohistochemical expression using NPC-1C Mab in colonic and pancreatic tissues (malignant and non-malignant). Around 53% of cells in a pancreatic cancer cell stained for NPC-1C by flow cytometry, and 48% (52/108) of pancreatic tissues stained positive with NPC-1C Mab by IHC. Similarly, 43% of colon cancer tissues were positive for NPC-1C by immunohistochemistry, while normal colon tissues were negative for NPC-1C. The in vivo anti-tumor activity of NPC-1C against pancreatic cancer was shown in a mouse model.

Notable takeaways from these two studies on NPC-1C-reactive in the context of PDA are: (a) it is expressed exclusively in pancreatic and colon cancer tissues; (b) it is different from native MUC5AC expressed in lung cancers and normal tissues identified by 45M1; (c) it is detected in the serum of pancreatic cancer patients; (d) it is immunogenic.

## 8. PAM4 Antibody Reactive MUC5AC Variant

In 1983, Gold et al. attempted to identify human pancreatic tissue-specific biomarkers in a study using mice, rabbits, and human tissues [79]. Pancreatic ductal mucin (PDM) was isolated from normal pancreatic tissues of autopsy specimens, and anti-PDM anti-sera was prepared by immunizing rabbits with PDM. Mucin-producing organs, including the pancreas, esophagus, stomach, lung, ovary, and breast, were stained using rabbit anti-PDM sera. The only two organ tissues that were positive with the antibody (after absorption by mucinous ovarian cystadenocarcinoma fluid) were the pancreas and goblet cells of the colon. Pancreas-specific determinants isolated from serum’s immunoelectrophoretic pattern (by separating the colonic mucin-specific determinants) were similar to those elicited after transplanting the pancreatic cancer xenografts in athymic nude mice.

This study led to the development of PAM4 antibody (Clivatuzumab tetraxetan), an immunoglobulin produced by immunization of mice with mucin purified from the xenografted RIP I human pancreatic carcinoma [80]. PAM4 antibody stained (via IHC) malignant tissues from pancreatic (85%), colon (50%), lung (5%), and stomach (20%) [80]. The normal tissues from the pancreas, lung, ovary, and spleen did not have reactivity to PAM4. The target for PAM4 antibody (PAM4-antigen) was later identified on MUC5AC, precisely e-fragment (AA1575-2052) comprising Cys2-3-4 on N-terminus and was different from the epitopes of commonly used anti-MUC5AC antibodies CLH2, 45-M1, and 2-11M1 Mabss [64,81]. In the pancreatic cancers, expression of PAM4 differed with the grade of differentiation of tumors (pronounced in well and moderately differentiated tissues and was absent in most poorly differentiated cancers) [80].

PAM-based serum enzyme immunoassays were also developed with a sensitivity of 82% and specificity of 95% in detecting PDA with a false-positive rate of 5% [82]. The test’s sensitivity and the antigen levels were higher in advanced cancers (Stage 3 or 4) compared to early stages (Stage 1 or 2). Even though 38% of chronic pancreatitis patients were positive, their corresponding tissues were negative for antibody (on IHC). The sensitivity dropped to 76% with PAM 4 assays in more extensive studies, but when combined with CA 19-9, the sensitivity improved back up to 84% [83].

The evidence available on MUC5AC reactive to NPC-1C and PAM4 antibody suggests that both the Mabs stain or react to a specific epitope (on MUC5AC) that is (a) not expressed in normal pancreatic tissues; (b) detected in most malignant pancreatic and colon tissues; (c) different from the targets of anti-MUC5AC monoclonal antibodies such as CLH2, 45-M1, and 2-11M used to detect native MUC5AC; and d) detected in the serum of pancreatic cancer patients (summarized in Table 3). NPC-1C and PAM4 antigen epitopes thus appear to represent specific changes on the native MUC5AC glycoprotein linked to a particular step in carcinogenesis. As with CLH2, neither of these Mabs were tested in non-malignant abnormal pancreatic tissues such as pancreatitis, hyperplasia, or PanIN. Further studies are needed to determine if they have better prognostic/predictive value compared to CLH2 Mab.

## 9. Future Directions with MUC5AC

We need more studies focusing on the mechanism that explains the de novo synthesis of mucins such as MUC5AC in pancreatic tissues that is constitutively expressed in other normal organs. In a current clinical practice where comprehensive genomic profiling is routine, IHC may be an outdated test. MUC5AC gene expression (mRNA) studies may give us more reliable data to do correlative studies in PDA. For IHC, investigating the expression patterns of MUC5AC detected by various Mabs (CLH1, 45M1, 21M1, NPC-1C, and PAM4) in PDA, associating it with the outcomes, and stratifying the data based on the stage of diagnosis, histology, differentiation, and thresholds for cut-offs might give us a better picture of its significance. In addition, examining the influence of MUC5AC on a patient’s response to therapeutic agents (gemcitabine-based vs. 5FU-based therapy) may help us choose appropriate systemic therapy.

Compared to pancreatic tissues, MUC5AC in lung and gastric tissues was studied extensively [84,85,86]. In the airways, its production is intricately related to Acinetobacter baumannii, and in the gastric lining, it helps H pylori colonialization, a known risk factor for gastric cancer [87,88]. It is worth investigating the influence of MUC5AC on the pancreas’s microenvironment and vice versa, and the impact of such influence on carcinogenesis.NPC-1C and PAM4 Mabs failed to have a meaningful impact in treating pancreatic and colon cancers [89,90,91,92,93,94,95]. There is preliminary evidence that MUC5AC is immunogenic, and the cytotoxic T lymphocytes (CTLs) are stimulated by it (peptides such as MUC5AC-A02-1398 [FLNDAGACV] and MUC5AC-A24-716 [TCQPTCRSL]) can kill the pancreatic tumor cells and have the potential to be harnessed for vaccine development [32]. MUC5AC expression in serum, pancreatic juice, and cyst fluid might help screen high-risk individuals for PDA, like young type 2 diabetes mellitus patients and patients with chronic pancreatitis.

## 10. Conclusions

There is a critical need to identify reliable biomarkers in the management of PDA. As of now, there is no valid clinical use for MUC5AC detected by IHC in PDA tissues in clinical practice. More clinical and pre-clinical studies are required to understand the impact of MUC5AC on the outcomes of pancreatic cancers. Specifically, the relevance of predicting outcomes based on expression patterns of MUC5AC detected by various Mabs (CLH1, 45M1, 21M1, NPC-1C, and PAM4) in PDA is unknown. Examining the influence of MUC5AC on patient response to therapeutic agents may aid in selecting systemic therapies in the future.

## Figures and Tables

**Figure 1 cancers-13-03059-f001:**
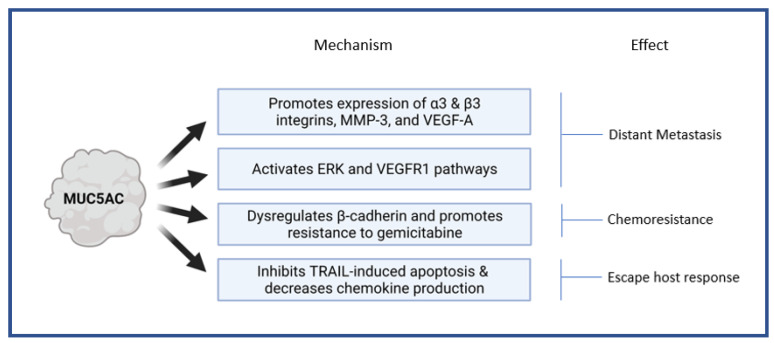
Pathophysiological mechanisms influenced by MUC5AC in pancreatic tissue. MMP-3—matrix metalloproteinase; VEGF—Vascular endothelial growth factor; VEGFR—Vascular endothelial growth factor receptor; ERK—Extracellular-signal-regulated kinase; TRAIL—TNF-related apoptosis-inducing ligand.

**Table 1 cancers-13-03059-t001:** Immunohistochemical expression of different mucins in pancreatic tissues [14,15].

Mucin	Normal Tissues	Pre-Cancerous	Malignant	IPMN/MCN	Benign Diseases
MUC1	E	E	E	E	E
MUC2	E	NE	E	E	E
MUC3	E	E	E	E	E
MUC4	NE	E ^1^	E ^1^	E ^1^	NE
MUC5AC	NE	E	E	E	E
MUC6	E	E	E	E	E

^1^ MUC4—expression is low; E—expressed; NE—not expressed; IPMN—intraductal papillary mucinous neoplasms; MCN—mucinous neoplasms.

**Table 2 cancers-13-03059-t002:** Previous studies in pancreatic cancer with MUC5AC and outcomes **.

Study (N)	Specimen Source	MUC5AC Variant Used for IHC	Expression Site—Cytoplasmic vs. Apical	Differentiation Expression Level Threshold—Positive vs. Negative	Positive MUC5AC vs. Negative MUC5AC Tumors (N)	Outcome in MUC5AC-Positive Tumors	Outcome
Takikita, 2009, All ductal adenocarcinomas (Well, moderate, and poor differentiated tumors) (154) [55].	Tissue microarrays	CLH2 monoclonal antibody **	No distinction	Negative/weak (negative) vs. moderate/strong (positive)	99 vs. 55	Better survival	Median OS, 10 vs. 5 m 1
Takikita, 2009, All cancers (adenocarcinoma +NET) (161) [55]					99 vs. 62	Worse survival	Median OS, 5 vs. 9.5 m 1
Takikita, 2009, Only well and moderate differentiated ductal adenocarcinomas (120) [55].					82 vs. 38	Worse Survival	Median OS 5 vs. 12 m 1
Higashi (114) [56]	EUS-FNA		Membrane (apical) or Cytoplasm	<10% vs. ≥10%	68 vs. 20	Better survival in Stage III/IV PDA	
Jinfeng, 2003 (33) [57]	Resection		Cytoplasm	<5% vs. ≥5%	21 vs. 12	Better survival	MUC5AC-negative expression was associated LN/LVI/AVI
Takano (59) [58]	Resection		No distinction	≤25% vs. >25%	15 vs. 44	More LN metastatic disease	
Aloysius * (104) [59]	Resection		Cytoplasmic	High (≥100) vs. low (<100) H-score	29 vs. 75	Worse survival Increased risk of recurrence ***	LVI/PNI ***

* In periampullary tumors, not specifically in pancreatic cancers. ** All studies used commercially available CLH2 monoclonal antibody for immunohistochemistry; *** Associated with high H-score; IHC—immunohistochemistry; PDA—pancreatic ductal adenocarcinoma; NET—neuroendocrine tumors; LN—lymph node; LVI—lymph vascular invasion; PNI—perineural invasion; EU—endoscopic ultrasound; FNA—fine needle aspirate; m—months; Mab—monoclonal antibody; 1—median OS of MUC5AC-positive vs. MUC5AC-negative tumors.

**Table 3 cancers-13-03059-t003:** NPC-1C and PAM4 monoclonal antibody in pancreatic ductal adenocarcinoma.

Characteristic	NPC-1C	PAM4
Target	Not defined	e-fragment (AA1575-2052) comprising Cys2-3-4 on N-terminus
PDA prevalence	48% by IHC 53% FC	85% by IHC
Serum testing in PDA (% above cut-off)	82% *	81% **
Prevalent in other cancer tissues	Colon ***	Colon, lung, and gastric
Similar characteristics	Negative in normal pancreatic tissue Targets are different from CLH2, 45-M1, and 21-1 mabs Immunogenic	

* 5/41 samples cancer patients tested with ELISA had pancreatic cancer, Cut-off value 355 cells/well; ** cut-off value -2.4 units/mL; *** Not present in lung cancer. IHC—immunohistochemistry; PDA—pancreatic ductal adenocarcinoma; mab—monoclonal antibody; FC—flow cytometry.

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
