# Peer review of "Understanding the Clinical Impact of MUC5AC Expression on Pancreatic Ductal Adenocarcinoma"

_cancers, 2021, doi:10.3390/cancers13123059_

Round 1
Reviewer 1 Report
This manuscript gathers old and recent data on MUC5AC in the context of PDAC. The review work is interesting and replaces MUC5AC as a potential promising PDAC biomarker. The authors should consider the following remarks to improve the review.
Line 45-57: not needed. To be deleted.
Line 59/60. « Carbohydrate antigen 19-9 (CA 19-9) is a blood-based molecular biomarker frequently used in PDA to diagnose, screen » Please rephrase this statement as this biomarker is NOT recommended as a diagnostic or screening tool for PDAC.
Line 63: “it is not elevated in patients without a Lewis antigen”. Specify the % of the general population concerned by this.
Line 97/98 “Among all the investigated mucins in pancreatic tissues, MUC5AC stands unique, as its mere detection in pancreatic tissues is abnormal and is expressed in high levels”. Is there missing words here (in PDAC or in advanced carcinomas) ?
Lines 117/11/ “When used along with CA 19-9, serum MUC5AC is useful in that it helps differentiate PDA from other benign conditions like chronic pancreatitis [43, 44]” please specify to what extent (sensitivity/specificity/accuracy).
Line 325: when stating “MUC5AC gene expression (mRNA) studies may give us more reliable data to do correlative studies in PDA”, are you sure that there is a strict correlation between RNA and protein levels (no posttranscriptional/translational/posttranslational regulations)?
Line 304: “PAM4 serum assays were also developed”. What kind of assays?
Lines 3035/336: “, it helps H pylori bind to the gastric epithelium and 335 help in carcinogenesis [91, 92]. Therefore, it is worthwhile to study the influence of 336 MUC5AC in shaping the microbiome conducive to pancreatic cancer development.” Please moderate the statement
Table 1: Expression can be determined by RT-QPCR and/or WB and/or immunohistochemistry. Please indicate for each gene the mode of quantification.
Please include in Table 2 the antibody used for MUC5AC quantification and the variant when relevant.
A graphical summary of MUC5AC native and tumor-related protein functions in PDAC physiopathology would be a big plus.
I think it is necessary to expend the review on the detection of MUC5AC on extracellular vesicles, in liquid biopsy, especially in the context for the search of biomarkers. Cite: Yang KS, Ciprani D, O'Shea A, Liss AS, Yang R, Fletcher-Mercaldo S, Mino-Kenudson M, Fernández-Del Castillo C, Weissleder R. Extracellular Vesicle Analysis Allows for Identification of Invasive IPMN. Gastroenterology. 2021 Mar;160(4):1345-1358.e11. doi: 10.1053/j.gastro.2020.11.046. Epub 2020 Dec 7. PMID: 33301777; PMCID: PMC7956058.
Reviewer 2 Report
The authors reviewed the clinical impact of MUC5AC expression on Pancreatic Ductal Adenocarcinoma. The applied threshold, detected region, and monoclonal antibodies were described well. It would be helpful for the readers studying markers for pancreatic ductal adenocarcinoma.
- Table 2: How about including the name of assays used in each studies. (commercially available assay? or in-house method?).
- Table 2: Aloysius ">25% vs. < 25%" should be corrected.
- Adding tables for NPC1 antibody and PAM4 antibody is recommended.
- Studies in Table 2 were published before 5 years ago. Adding recent studies and describing changes are necessary.
Round 2
Reviewer 2 Report
The comments for improving manuscript are addressed.